# Light Interaction with Cluster Chiral Nanostructures by High-Order Bessel Beam

**Jing Bai** [1], **Cheng-Xian Ge** [2], **Zhen-Sen Wu** [3], **Peng Su** [1,*] and **Yu Gao** [1]

[1] School of Electronic Engineering, Xi'an University of Posts & Telecommunications, Xi'an 710121, China; jbai@stu.xidian.edu.cn (J.B.); 1977574972@stu.xupt.edu.cn (Y.G.)
[2] The 39th Research Institute of China Electronics Technology Corporation, Xi'an 710065, China; zhaoxu@xidian.edu.cn
[3] School of Physics and Optoelectronic Engineering, Xidian University, Xi'an 710071, China; cxge@stu.xidian.edu.cn
[*] Correspondence: supeng1223@stu.xupt.edu.cn; Tel.: +86-134-7405-9068

**Abstract:** Interactions between cluster chiral nanoparticles and a high-order Bessel beam (HOBB) with arbitrary illuminations are investigated. The generalized Lorenz–Mie theory (GLMT) is applied to derive the expansions of HOBB. Based on the additional theorem, multiple scattering results of cluster chiral nanoparticles are obtained by taking into account the tangential continuous boundary conditions. The present theory and codes proved to be effective when confronted with the simulations obtained from the Computer Simulation Technology (CST) software. Numerical results concerning the effects of beam order, beam conical angle, incident angles, beam polarization state, the chirality, and the material loss on the scattering of various types of aggregated chiral particles are displayed in detail, including the linearly chiral sphere chain, the chiral cube array, and the complex models composed of aggregated chiral spheres. This study may provide critical support to analytically understand the optical scattering characteristics with aggregated chiral particles of complex shapes, and may find important applications in manipulating collective chiral particles.

**Keywords:** multiple scattering; collective chiral particles; high-order Bessel beam; spherical vector wave functions

## 1. Introduction

Chiral media are important subjects for researchers due to their unique properties and their wide applications in many fields, including physical detection, biological manipulation, communication, and particle sizing. In the microwave region, researchers have fabricated chiral materials by randomly integrating micro-chiral objects, such as metal spirals, into ordinary dielectrics [1].

Considering the increasing applications of chiral materials in biology and technology, the interactions of chiral materials with electromagnetic fields have recently become a very interesting research subject. In the past few decades, a large number of scholars have conducted a lot of analytical research on the interactions between chiral media and electromagnetic waves. In 1972, Gordon first proposed the scattering problem of chiral dielectric spheres based on the classical Mie scattering theory [2]. In 1974, Bohren analytically obtained the classical scattering solutions of chiral particles [3]. Subsequently, studies of chiral spherical scattering have been expanded to different aspects, including scattering by an infinitely long chiral cylinder [4], the scattering problem of chiral particles implanted in mediators [5], scattering of ellipsoids with chirality [6], and scattering of non-uniformly distributed chiral spheres [7]. Nevertheless, the number of scatterers in the above investigations is limited to a single chiral object, and the analytical study of interactions between aggregated chiral objects is very few.

The study of scattering by collective objects is important due to their comprehensively practical utilizations. Since Mackowski [8] and Xu [9] promoted and developed the additional theorem of spherical vector wave functions (SVWFs), the compact analysis for the multi-scattering of aggregated isotropic spherical particles have been derived and widely developed. Later, the scattering of multiple spheres was extended from the former linear system to the more complex case of cluster aggregation by Fuller and Kattawar [10]. Xu [11] proposed the generalized multiple Mie (GMM) theory to obtain the multi-scattering characteristics of assembly isotropic spheres by plane waves incidence. Based on GMM theory, Gouesbet et al. [12,13] developed the analytical solution of scattering by multiple spheres with an arbitrary profiled beam incidence, involving many improvements of components developed from the GLMT [14–16]. Since then, by applying the method of Gouesbet, many subsequent studies such as soot collectivity [17] and periodic arrays [18] for the scattering characteristics of multiple particles have been comprehensively reported. Moreover, some numerical approaches including the null-field theory [19] and dipolar approximation [20] have also been efficient methods to solve the problem. Despite the extensive knowledge gained from these investigations, previous studies generally focused on the interactions between Gaussian beam or plane wave incidence for assembly isotropic particles.

Since Durnin first introduced the concept of the Bessel beam, the unique beam has attracted increasing attention due to its special properties of non-diffraction and self-reconstruction [21,22]. It appears to be a potential dramatic alternative to using Gaussian beams in some particular scenarios [23,24]. Driven by the characteristics, a variety of researches have been proposed to study the description, expansion, scattering, and applications in the fields of capture, communication, and detection [25,26]. Recently, a convenient technique for realizing the orientation and topological charge of HOBB has been reported by Dwivedi, which provides significant advantages in drilling applications [27,28]. Moreover, some analytical studies have been carried out on the description of HOBB by a double integration in a sphere coordinate system [29–31]. Taking into account the complex and time-consuming process of numerical calculation [32], several studies have been involved in the accurate investigation of expansions by using the conveniently analytical methods [33–35]. Specially, the angular spectrum representation (ASR) is promoted by Lock [36] and applied to the general description of Bessel beam with zero-order. Based on the same method, Ma et al. researched isotropic sphere scattering by Bessel beam with un-polarizations [37]. In addition, Gouesbet and Lock developed the dark theorem to describe the BSCs and verified the presence of Bessel beams with no vortex [38,39]. Wang deduced the expansion of HOBB with circular symmetry distribution [40,41].

Furthermore, from the relevant circumstances, the investigations of HOBB scattering [42,43] by chiral particles [39], by anisotropic particles [44], and by multi-layered particles [45] are researched widely. However, the work mentioned above is mainly devoted to the interactions of the individual object. Studying the scattering of several spheres at the same time is very different to that of a single sphere, particularly for the case of aggregated chiral particles, since the particles encountered in nature or manufacturing are often found to be clustered. Accurate prediction of scattering of multiple chiral spheres simultaneously with an arbitrary incident HOBB by using a rigorous analytical solution may have some significance in controlling and manipulating collective chiral particles of complex shapes operating with a non-diffracting beam. In particular, using the analytical methods can provide more physical insights into the problem and obtain accurate results which can be used in comparison with numerical methods.

The body of the thesis is ordered as follows. In Section 2, based on the framework of GLMTs, the expressions of arbitrary polarized HOBB are given. Moreover, analytical solutions of the scattering interactions between collective chiral nanoparticles and arbitrarily illuminated HOBB are presented by using the GMM equation. Section 3 presents the numerical simulations for different parameters. Finally, Section 4 investigates a conclusion.

## 2. Theoretical Analysis

The geometry of the multiple chiral particles system induced by a polarized Bessel beam is given in Figure 1. The particle system $O_j x_j y_j z_j$ is defined coincident with the fixed global coordinate $Oxyz$. The particles are arbitrarily induced by a HOBB. A temporary coordinate system $O_j x_j' y_j' z_j'(j = 1, 2)$ is depicted parallel to the beam system $O'x'y'z'$. The incident angle between propagation direction and $z$ axis is represented as $\alpha$, and we denote the polarized $\beta$ as an angle of electric vector vibrating direction of HOBB. When the case is vertical incidence, the electric field vibrates in the lateral magnetic pattern (TM) with $\beta = 0°$; vice versa, the magnetic field vibrates in the lateral electric pattern (TE) with $\beta = 90°$. When $\beta$ presents other values, it represents another polarization mode.

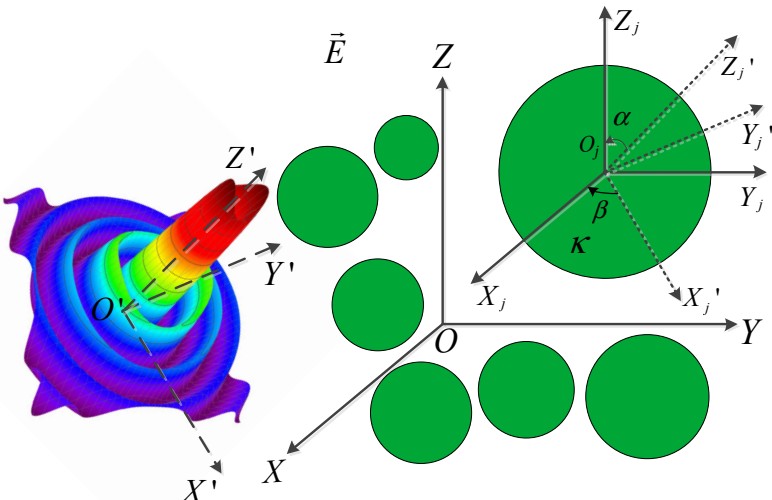

**Figure 1.** Illustration of multiple chiral spheres scattering induced by an arbitrarily polarized high-order Bessel bam (HOBB).

To study the scattering interactions of HOBB and multiple particles, it is necessary to come up with a suitable Bessel beam expansion. In 2017, Wang [36] derived a general expansion of HOBB which fits well with the equations of Maxwell. Obviously, the descriptions of Bessel beams can also be applied to our studies. On the basis of this work, the initial $x$-polarized HOBB in the system $Oxyz$ is described as [40,46]:

$$
\begin{aligned}
\boldsymbol{E} = {} & E_0 g(\alpha_0)(-i)^p e^{ip\varphi} \exp[-ik_z(z - z')] \\
& \times \left\{ \left[ (1 + \cos\alpha_0)J_p(k_t\rho) + \tfrac{1-\cos\alpha_0}{2}[e^{2i\varphi}J_{p+2}(k_t\rho)] + e^{-2i\varphi}J_{p-2}(k_t\rho) \right] \hat{e}_x \right. \\
& + \left[ \tfrac{1}{2i}(1 - \cos\alpha_0)[e^{2i\varphi}J_{p+2}(k_t\rho)] - e^{-2i\varphi}J_{p-2}(k_t\rho) \right] \hat{e}_y \\
& \left. + \left[ i\sin\alpha_0[e^{i\varphi}J_{p+1}(k_t\rho)] - e^{-i\varphi}J_{p-1}(k_t\rho) \right] \hat{e}_z \right\}
\end{aligned}
\tag{1}
$$

in which $E_0$ and $\alpha_0$ are the electric field magnitude and the half-conical angle, respectively. $k_z = k\cos\alpha_0$, $k_t = k\sin\alpha_0$. The general function $g(\alpha_0)$ can make the expansion of Equation (1) degenerate into a Davis HOBB proposed in [36,47] if $g(\alpha_0) = (1 + \cos\alpha_0)/4$. Moreover, they degenerate to a partial wave HOBB developed in [48] if $g(\alpha_0) = 1/2$.

Considering that the mathematical expression of Equation (1) originates from equations of Maxwell, the $x$-polarized HOBB can be expanded on the basis of SVWFs in the intermediate system $O_j x_j' y_j' z_j'$ as [46]:

$$
\begin{aligned}
\boldsymbol{E}_j^{inc} &= E_0 \sum_{n=1}^{\infty} \sum_{m=-n}^{n} C_{nm} \left[ -i g'^{m}_{jn,TE} \boldsymbol{M}^{(1)}_{mn}(\boldsymbol{r}_j', k) + g'^{m}_{jn,TM} \boldsymbol{N}^{(1)}_{mn}(\boldsymbol{r}_j', k) \right] \\
\boldsymbol{H}_j^{inc} &= E_0 \frac{k_0}{\omega\mu_0} \sum_{n=1}^{\infty} \sum_{m=-n}^{n} C_{nm} \left[ g'^{m}_{jn,TE} \boldsymbol{N}^{(1)}_{mn}(\boldsymbol{r}_j', k) + i g'^{m}_{jn,TM} \boldsymbol{M}^{(1)}_{mn}(\boldsymbol{r}_j', k) \right]
\end{aligned}
\tag{2}
$$

where the notation "*inc*" shows the relevant incident expressions, and the subscript indicates the relevant *j*th spherical expansions. $M_{mn}^{(q)}(r,k)$ and $N_{mn}^{(q)}(r,k)$ are the SVWFs, where $q = 1, 2, 3, 4$ represents four kinds of spherical Bessel functions in the SVWFs.

$$C_{nm} = kC_n^{PW}(-1)^{(m-|m|)/2}\frac{(n-m)!}{(n+|m|)!} \quad C_n^{PW} = (-i)^{n+1} \cdot 2n + \frac{1}{kn(n+1)} \tag{3}$$

According to the derivation obtained in GLMT [21], the expression of $g'^m_{jn,TE}$ can be written as follows [46]:

$$
\begin{aligned}
g'^m_{jn,TE} = \quad & ig(\alpha_0)(-1)^{(m-|m|)/2}\frac{(n-m)!}{(n+|m|)!}\exp(ik_z z_j') \\
& \{i^{p-m+1}e^{(p-m+1)\phi_j'}J_{p-m+1}(k\rho_j'\sin\alpha_0)[\tau_n^m(\cos\alpha_0) + m\pi_n^m(\cos\alpha_0)] \\
& -i^{p-m+1}e^{(p-m-1)\phi_j'}J_{p-m-1}(k\rho_j'\sin\alpha_0)[\tau_n^m(\cos\alpha_0) - m\pi_n^m(\cos\alpha_0)]
\end{aligned}
\tag{4}
$$

in which, $\tau_n^m(\cos\alpha_0) = dP_n^m(\cos\alpha_0)/d\alpha_0$, $\pi_n^m(\cos\alpha_0) = P_n^m(\cos\alpha_0)/\sin\alpha_0$, $p$ denotes the beam order and

$$\rho_j' = \left[(x_j')^2 + (y_j')^2\right]^{1/2} \quad \phi_j' = \tan^{-1}\left(\frac{y_j'}{x_j'}\right) \tag{5}$$

Due to the coordinate rotation theorem of SVWFs [49], the formulations of SVWFs between the coordinates $O_j x_j' y_j' z_j'$ and $O_j x_j y_j z_j$ have the following relations:

$$(M, N)_{mn}^{(1)}(r_j', k) = \sum_{s=-n}^{n} \chi(m, s, n)(M, N)_{sn}^{(1)}(r_j, k) \tag{6}$$

in which, the beam center $O'$ coordinate $(x_j', y_j', z_j')$ at the temporary system $O_j x_j' y_j' z_j'$ is derived as

$$
\begin{pmatrix} x_j' \\ y_j' \\ z_j' \end{pmatrix} = \begin{pmatrix} \cos\alpha\cos\beta & -\sin\beta & \sin\alpha\cos\beta \\ \cos\alpha\sin\beta & \cos\beta & \sin\alpha\sin\beta \\ -\sin\alpha & 0 & \cos\alpha \end{pmatrix} \begin{pmatrix} x' - x_j \\ y' - y_j \\ z' - z_j \end{pmatrix}
\tag{7}
$$

In practice, we need to gain the expansion of HOBB with arbitrary incident directions in the chiral sphere system $(O_j x_j y_j z_j)$. By substituting Equation (6) into Equation (2), the electromagnetic fields of arbitrarily incident HOBB can be expanded in the *j*th chiral system $O_j x_j y_j z_j$ as:

$$
\begin{aligned}
E_j^{ip} &= E_0 \sum_{n=1}^{\infty} \sum_{m=-n}^{n} \left[a_{jmn}^{ip}M_{mn}^{(1)}(r_j, k) + b_{jmn}^{ip}N_{mn}^{(1)}(r_j, k)\right] \\
H_j^{ip} &= E_0 \frac{k_0}{\omega\mu_0} \sum_{n=1}^{\infty} \sum_{m=-n}^{n} \left[a_{jmn}^{ip}N_{mn}^{(1)}(r_j, k) + b_{jmn}^{ip}M_{mn}^{(1)}(r_j, k)\right]
\end{aligned}
\tag{8}
$$

where

$$\begin{pmatrix} a_{jmn}^{ip} \\ b_{jmn}^{ip} \end{pmatrix} = \sum_{s=-n}^{n} \chi(s, m, n)C_{ns}(-ig'^m_{n,TE}, g'^m_{n,TM}) \tag{9}$$

$$\chi(s, m, n) = (-1)^{m+s}\left[\frac{(n+m)!(n-m)!}{(n+s)!(n-s)!}\right]^{1/2}u_{ms}^{(n)}(-\alpha) \tag{10}$$

$$u_{ms}^{(n)}(-\alpha) = \sum_{\sigma=max(0,-s-m)}^{min(n-m,n-s)} (-1)^{n-m-\sigma}\begin{pmatrix} n+s \\ n-m-\sigma \end{pmatrix}\begin{pmatrix} n-s \\ \sigma \end{pmatrix}\left(\cos\frac{\alpha}{2}\right)^{2\sigma+m+s}\left(\sin\frac{-\alpha}{2}\right)^{2n-2\sigma-m-s} \tag{11}$$

where $a_{jmn}^{ip}$ and $b_{jmn}^{ip}$ indicate the incident beam coefficients, superscript *ip* corresponds to different polarization states, such as *ix*, *iy*, *iL*, and *iR* modes, respectively. Relations between *y*-polarization HOBB and *x*-polarization HOBB can be expressed as: $a_{jmn}^{iy} = -ib_{jmn}^{ix}$,

$b_{jmn}^{iy} = -ia_{jmn}^{ix}$. In addition, the coefficients of a right-circular polarized (RCP) and left-circular polarized (LCP) Bessel beam can be obtained, respectively, as follows:

$$a_{jmn}^{iR} = \sqrt{2}(a_{jmn}^{ix} + b_{jmn}^{ix})/2 \quad b_{jmn}^{iR} = \sqrt{2}(b_{jmn}^{ix} + a_{jmn}^{ix})/2$$
$$a_{jmn}^{iL} = \sqrt{2}(a_{jmn}^{ix} - b_{jmn}^{ix})/2 \quad b_{jmn}^{iL} = \sqrt{2}(b_{jmn}^{ix} - a_{jmn}^{ix})/2 \tag{12}$$

To give the readers a more intuitive representation, the magnitude plots for the electric magnitude of a HOBB with $x$-polarization, as well as the $y$-polarization, the RCP, and the LCP polarization, are given in Figure 2. All the beams are denoted by $p = 2$ and $\alpha_0 = 30°$, as well as the wavelength $\lambda = 1064nm$. As can be seen, the beam intensity distribution exhibits good symmetry in different polarization modes. Moreover, the typical bright, circular core surrounded by concentric rings of approximately the same power is visible. One can find that, as the distance from the center increases, the max of the intensity decreases and the intensity in the inner circle is larger. In addition, all high-order Bessel beams have zero central amplitude, as shown in Figure 2a–p; this can be explained by the fact that a HOBB propagates over a characteristic length without spreading (dark central region) and the doughnut shape is conserved. Based on the focusing properties, one can expect that these beams can be used to guide high-index or low-index particles along parabolic trajectories. In this paper, we are concerned with chiral nano-particles scattering in the focal region.

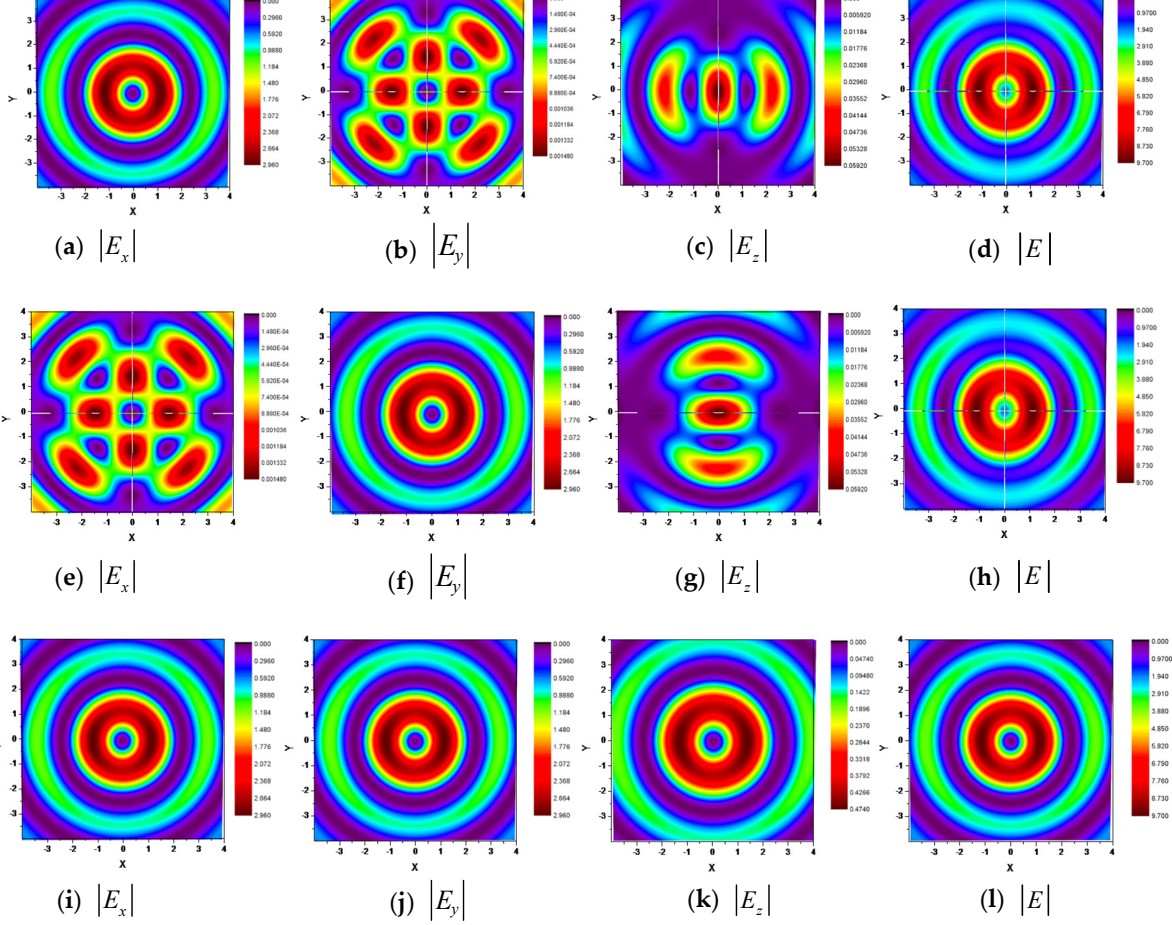

(a) $|E_x|$     (b) $|E_y|$     (c) $|E_z|$     (d) $|E|$

(e) $|E_x|$     (f) $|E_y|$     (g) $|E_z|$     (h) $|E|$

(i) $|E_x|$     (j) $|E_y|$     (k) $|E_z|$     (l) $|E|$

**Figure 2.** *Cont.*

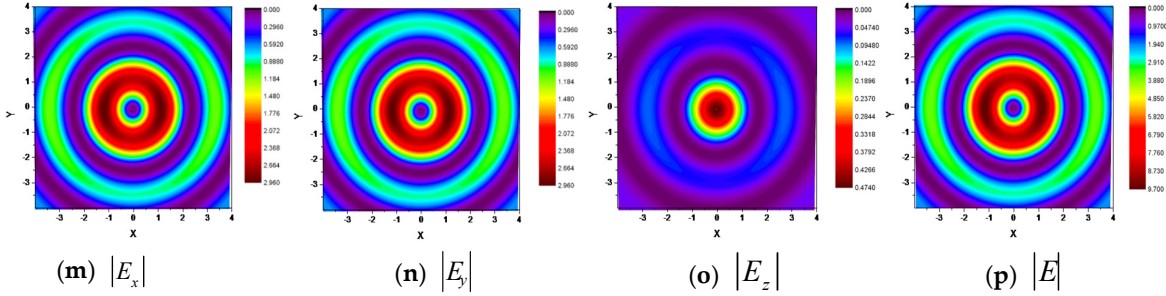

(**m**) $|E_x|$      (**n**) $|E_y|$      (**o**) $|E_z|$      (**p**) $|E|$

**Figure 2.** Illustration of the electric components of arbitrarily polarized HOBB. (**a–d**) x-polarized; (**e–h**) y-polarized; (**i–l**) RCP mode; (**m–p**) LCP mode.

### 3. Multi-Scattering Theory

Taking into account the eigen-mode in chiral media made up of two parts: the RCP wave with $k_1 = \omega(\sqrt{\mu_c \varepsilon_c} + \kappa\sqrt{\mu_0 \varepsilon_0})$ and LCP wave with $k_2 = \omega(\sqrt{\mu_c \varepsilon_c} - \kappa\sqrt{\mu_0 \varepsilon_0})$; where $\varepsilon_c$, $\mu_c$, and $\kappa$ represent the permittivity, permeability, and chirality parameter of chiral sphere, and $\varepsilon_0$ and $\mu_0$ denote permittivity and permeability in vacuum, the corresponding electromagnetic fields of the $j$th particle are described as [3]:

$$E_j^s = \sum_{n=1}^{\infty} \sum_{m=-n}^{n} E_0 \left[ a_{jmn}^s \boldsymbol{M}_{mn}^{(3)}(\boldsymbol{r}_j, k) + b_{jmn}^s \boldsymbol{N}_{mn}^{(3)}(\boldsymbol{r}_j, k) \right]$$
$$H_j^s = \frac{k_0}{\omega \mu_0} \sum_{n=1}^{\infty} \sum_{m=-n}^{n} E_0 \left[ a_{jmn}^s \boldsymbol{N}_{mn}^{(3)}(\boldsymbol{r}_j, k) + b_{jmn}^s \boldsymbol{M}_{mn}^{(3)}(\boldsymbol{r}_j, k) \right]$$

(13)

$$E_j^1 = \sum_{n=1}^{\infty} \sum_{m=-n}^{n} \left[ A_{jmn}^1 \boldsymbol{M}_{mn}^{(1)}(\boldsymbol{r}, k_1) + A_{jmn}^1 \boldsymbol{N}_{mn}^{(1)}(\boldsymbol{r}, k_1) + B_{jmn}^1 \boldsymbol{M}_{mn}^{(1)}(\boldsymbol{r}, k_2) - B_{jmn}^1 \boldsymbol{N}_{mn}^{(1)}(\boldsymbol{r}, k_2) \right]$$
$$H_j^1 = -i\sqrt{\varepsilon_c/\mu_c} \sum_{n=1}^{\infty} \sum_{m=-n}^{n} [ A_{jmn}^1 \boldsymbol{N}_{mn}^{(1)}(\boldsymbol{r}, k_1) + A_{jmn}^1 \boldsymbol{M}_{mn}^{(1)}(\boldsymbol{r}, k_1) + B_{jmn}^1 \boldsymbol{N}_{mn}^{(1)}(\boldsymbol{r}, k_2) - B_{jmn}^1 \boldsymbol{M}_{mn}^{(1)}(\boldsymbol{r}, k_2) ]$$

(14)

where the notations "1" and "$s$" represent the relevant internal and scattered parameters, respectively; $a_{jmn}^s$, $b_{jmn}^s$ indicate the scattering coefficients of the $j$th chiral particle; and $A_{jmn}^1$, $B_{jmn}^1$ denote the internal factors.

Following the continuous bounding criteria, the equations in the $j$th chiral particle system accord with:

$$E_j^1 \Big|_t = E_j^{it} \Big|_t + E_j^s \Big|_t, \quad H_j^1 \Big|_t = H_j^{it} \Big|_t + H_j^s \Big|_t$$

(15)

where $E_j^{it}$ and $H_j^{it}$ indicate the entire fields of the particle.

They can usually decompose into the initial fields and the fields scattered from another sphere $p$.

$$E_j^{it} = E_j^{ip} + \sum_{\substack{p=1 \\ (p \neq j)}}^{L} E_{p,j}^s \quad H_j^{it} = H_j^{ip} + \sum_{\substack{p=1 \\ (p \neq j)}}^{L} H_{p,j}^s$$

(16)

where $E_j^{ip}$ and $H_j^{ip}$ represent the initial beam; and $E_{p,j}^s$ and $H_{p,j}^s$ represent the re-scattered beam from the $p$th particle.

Applying the additional theorem [9] and Equations (8), (13), and (16), the entire electromagnetic fields at the $j$th sphere are:

$$E_j^{it} = E_0 \sum_{n=1}^{\infty} \sum_{m=-n}^{n} \left[ a_{jmn}^{it} \boldsymbol{M}_{mn}^{(1)}(\boldsymbol{r}_j, k) + b_{jmn}^{it} \boldsymbol{N}_{mn}^{(1)}(\boldsymbol{r}_j, k) \right]$$
$$H_j^{it} = E_0 \frac{k}{\omega \mu} \sum_{n=1}^{\infty} \sum_{m=-n}^{n} \left[ a_{jmn}^{it} \boldsymbol{N}_{mn}^{(1)}(\boldsymbol{r}_j, k) + b_{jmn}^{it} \boldsymbol{M}_{mn}^{(1)}(\boldsymbol{r}_j, k) \right]$$

(17)

The relevant coefficients are:

$$a_{jmn}^{it} = a_{jmn}^{i} + \sum_{\substack{(p \neq j)}}^{L} \sum_{v=1}^{\infty} \sum_{\mu=-v}^{v} \left[ a_{p\mu v}^{s} A_{mn}^{\mu v} + b_{p\mu v}^{s} B_{mn}^{\mu v} \right] (p \neq j)$$

$$b_{jmn}^{it} = b_{jmn}^{i} + \sum_{\substack{(p \neq j)}}^{L} \sum_{v=1}^{\infty} \sum_{\mu=-v}^{v} \left[ a_{p\mu v}^{s} B_{mn}^{\mu v} + b_{p\mu v}^{s} A_{mn}^{\mu v} \right] (p \neq j)$$

(18)

where $A_{\mu v}^{mn}$ and $B_{\mu v}^{mn}$ indicate the vector additional coefficients [9]. Replacing the boundary conditions in Equation (15) with Equations (13), (14), and (17), and using the expansions of SVWFs [50], the scattering coefficients of the $j$th chiral particle can be gained as:

$$a_{jmn}^{s} = a_{jmn} \left\{ a_{jmn}^{ip} + \sum_{\substack{(i \neq j)}}^{M} \sum_{v=1}^{\infty} \sum_{\mu=-v}^{v} \left[ a_{i\mu v}^{s} A_{mn}^{\mu v} + b_{i\mu v}^{s} B_{mn}^{\mu v} \right] (i \neq j) \right\}$$

$$b_{jmn}^{s} = b_{jmn} \left\{ b_{jmn}^{ip} + \sum_{\substack{(i \neq j)}}^{M} \sum_{v=1}^{\infty} \sum_{\mu=-v}^{v} \left[ a_{i\mu v}^{s} B_{mn}^{\mu v} + b_{i\mu v}^{s} A_{mn}^{\mu v} \right] (i \neq j) \right\}$$

(19)

in which $a_{jmn}$ and $b_{jmn}$ denote the scattered result of a single chiral sphere [51].

$$a_{jmn} = A_{jn}^{sa} a_{jmn}^{ip} + A_{jn}^{sb} b_{jmn}^{ip} \quad b_{jmn} = B_{jn}^{sa} a_{jmn}^{ip} + B_{jn}^{sb} b_{jmn}^{ip}$$

(20)

where

$$A_{jn}^{sa} = \frac{\psi_n(x_{0j})}{\xi_n(x_{0j})} \frac{\dfrac{D_n^{(1)}(x_{1j}) - \eta_r D_n^{(1)}(x_{0j})}{\eta_r D_n^{(1)}(x_{1j}) - D_n^{(3)}(x_{0j})} + \dfrac{D_n^{(1)}(x_{2j}) - \eta_r D_n^{(1)}(x_{0j})}{\eta_r D_n^{(1)}(x_{2j}) - D_n^{(3)}(x_{0j})}}{\dfrac{\eta_r D_n^{(3)}(x_{0j}) - D_n^{(1)}(x_{1j})}{\eta_r D_n^{(1)}(x_{1j}) - D_n^{(3)}(x_{0j})} + \dfrac{\eta_r D_n^{(3)}(x_{0j}) - D_n^{(1)}(x_{2j})}{\eta_r D_n^{(1)}(x_{2j}) - D_n^{(3)}(x_{0j})}}$$

(21)

$$A_{jn}^{sb} = \frac{\psi_n(x_{0j})}{\xi_n(x_{0j})} \frac{\dfrac{\eta_r D_n^{(1)}(x_{1j}) - D_n^{(1)}(x_{0j})}{\eta_r D_n^{(1)}(x_{1j}) - D_n^{(3)}(x_{0j})} - \dfrac{\eta_r D_n^{(1)}(x_{2j}) - D_n^{(1)}(x_{0j})}{\eta_r D_n^{(1)}(x_{2j}) - D_n^{(3)}(x_{0j})}}{\dfrac{\eta_r D_n^{(3)}(x_{0j}) - D_n^{(1)}(x_{1j})}{\eta_r D_n^{(1)}(x_{1j}) - D_n^{(3)}(x_{0j})} + \dfrac{\eta_r D_n^{(3)}(x_{0j}) - D_n^{(1)}(x_{2j})}{\eta_r D_n^{(1)}(x_{2j}) - D_n^{(3)}(x_{0j})}}$$

(22)

$$B_{jn}^{sa} = A_{jn}^{sb}$$

(23)

$$B_{jn}^{sb} = \frac{\psi_n(x_{0j})}{\xi_n(x_{0j})} \frac{\dfrac{\eta_r D_n^{(1)}(x_{1j}) - D_n^{(1)}(x_{0j})}{D_n^{(1)}(x_{1j}) - \eta_r D_n^{(3)}(x_{0j})} + \dfrac{\eta_r D_n^{(1)}(x_{2j}) - D_n^{(1)}(x_{0j})}{D_n^{(1)}(x_{2j}) - \eta_r D_n^{(3)}(x_{0j})}}{\dfrac{D_n^{(3)}(x_{0j}) - \eta_r D_n^{(1)}(x_{1j})}{D_n^{(1)}(x_{1j}) - \eta_r D_n^{(3)}(x_{0j})} + \dfrac{D_n^{(3)}(x_{0j}) - \eta_r D_n^{(1)}(x_{2j})}{D_n^{(1)}(x_{2j}) - \eta_r D_n^{(3)}(x_{0j})}}$$

(24)

where the expressions of $\psi_n(z)$, $\xi_n(z)$, and its logarithmic derivatives $D_n^{(1)} D_n^{(3)}$ can be referred from the existing references [51]. The other symbols represent $x_{0j} = k_0 a_j$, $x_{1j} = k_{1j} a_j$, $x_{2j} = k_{2j} a_j$, and $\eta_{rj} = \sqrt{\varepsilon_0 / \mu_0} / \sqrt{\varepsilon_{cj} / \mu_{cj}}$, where $k_0$ denote the surrounding vacuum wave number, and $k_{1j}$ and $k_{2j}$ pertain to the RCP and LCP wave at the $j$th sphere.

By utilizing the above scattering coefficients derived, the total scattered fields in Oxyz can be deduced, which consist of individual fields scattered from each chiral sphere:

$$E^{st} = \sum_{n=1}^{\infty} \sum_{m=-n}^{n} E_{mn} \left[ a_{mn}^{st} M_{mn}^{(3)}(\boldsymbol{r}_1, k_0) + b_{mn}^{st} N_{mn}^{(3)}(\boldsymbol{r}_1, k_0) \right]$$

$$H^{st} = \frac{k_0}{i\omega\mu_0} \sum_{n=1}^{\infty} \sum_{m=-n}^{n} \widetilde{B}_{\mu v}^{mn} \left[ a_{mn}^{st} N_{mn}^{(3)}(\boldsymbol{r}_1, k_0) + b_{mn}^{st} M_{mn}^{(3)}(\boldsymbol{r}_1, k_0) \right]$$

(25)

where $a_{mn}^{st}$ and $b_{mn}^{st}$ denote the entire scattered coefficients:

$$
\begin{aligned}
a_{mn}^{st} &= a_{1mn}^s + \sum_{v=1}^{\infty} \sum_{\mu=-v}^{v} \left[ a_{2\mu v}^s \widetilde{A}_{mn}^{\mu v}{}'(2,1) + b_{2mn}^s \widetilde{B}_{mn}^{\mu v}{}'(2,1) \right] \\
b_{mn}^{st} &= b_{1mn}^s + \sum_{v=1}^{\infty} \sum_{\mu=-v}^{v} \left[ a_{2\mu v}^s \widetilde{B}_{mn}^{\mu v}{}'(2,1) + b_{2mn}^s \widetilde{A}_{mn}^{\mu v}{}'(2,1) \right]
\end{aligned}
\tag{26}
$$

Utilizing the total scattering coefficients gained, we can get the radar cross section (RCS), which is presented as:

$$
\begin{aligned}
\sigma &= \lim_{r \to \infty} \left( 4\pi r^2 \left| E^{st} \right|^2 / \left| E^{ip} \right|^2 \right) \\
&= \frac{4\pi}{k_0^2} \left\{ \left| \sum_{n=1}^{\infty} \sum_{m=-n}^{n} E_{mn}(-i)^n e^{im\phi} \left[ m a_{mn}^{st} \frac{P_n^m(\cos\theta)}{\sin\theta} + b_{mn}^{st} \frac{dP_n^m(\cos\theta)}{d\theta} \right] \right|^2 \right. \\
&\quad \left. + \left| \sum_{n=1}^{\infty} \sum_{m=-n}^{n} E_{mn}(-i)^{n+1} e^{im\phi} \left[ a_{mn}^{st} \frac{dP_n^m(\cos\theta)}{d\theta} + m b_{mn}^{st} \frac{P_n^m(\cos\theta)}{\sin\theta} \right] \right|^2 \right\}
\end{aligned}
\tag{27}
$$

where $E^{ip}$ denotes the initial electric field with the amplitude presented as a unit.

### 4. Numerical Simulation

The scattering consequence of assembled chiral spheres illuminated by an arbitrarily polarized HOBB are numerically calculated in this part. The E-plane and H-plane correspond to the $xoz-$ plane and H-plane, respectively. Numerical simulations considering the various effects of different parameters including beam order, beam polarizations, incident angles, beam conical angle, the chirality, the material loss, the number of linear chain, as well as the periodical structure with dense chiral spheres on angular distributions of the RCSs are analyzed numerically in detail. The correctness of the present computation was confirmed when confronted with the numerical consequence obtained from the CST simulation. As shown in Figure 3, the structure of two identical chiral spheres are arranged symmetrically in the $\hat{z}$ axis by HOBB incidence in the $+\hat{z}$ direction. We present the incident beam with $x$ polarization and $y$ polarization in Figure 3a, b separately, where wavelength is defined as $\lambda = 0.6328 \ \mu\text{m}$. As verification, a HOBB will be degenerated to compare with the consequence simulated from CST. In the below plots, the well consistency can indicate the credibility of our result.

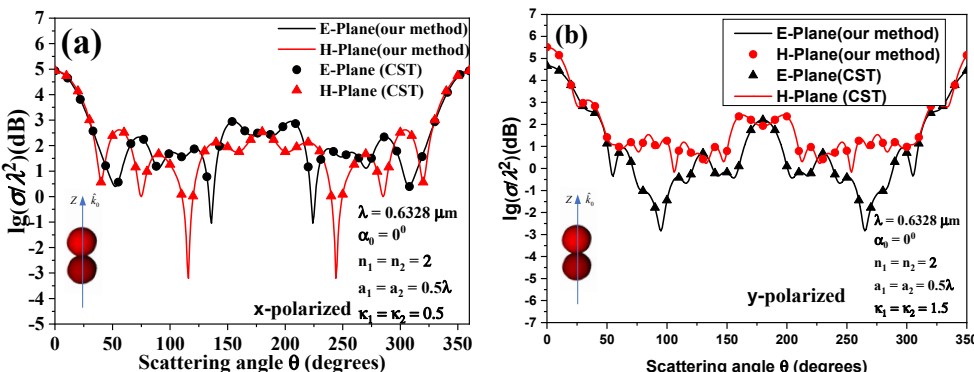

**Figure 3.** Contrast of our simulations of angular distributions of normalized RCS with that of CST software: (**a**) $x$-polarized; (**b**) $y$-polarized.

Figure 4 presents the RCSs distributions of a linear chain composed of three chiral particles by a HOBB disseminating in the angle of $30°$ from the $+ Z$ axis with different polarization states. The three close-packed chiral spheres are supposed to be the same, with $\kappa_j = 0.5$, $a_j = 0.5\lambda$, and $\lambda = 0.6328 \ \mu\text{m}$. For the calculation parameters, the refractive indices of the chiral mediums and outer space are $n_j = 2n_0$ and $n_0 = 1$. In addition, the beam

center, the second chiral sphere center, and the global system $Oxyz$ origin are coincident. The parameters ix, iy, iR and iL represent the states of linear polarizations along the *x*-axis and *y*-axis, and circular polarizations of right-handed and left-handed, respectively. As shown in Figure 4a,b, the maximum RCS occurs in the angle consisting of the incident angle and conical angle $\alpha_0 = 30°$. Indeed, for the considered polarizations, the amplitude of RCS oscillations is greater for a first-order Bessel beam with RCP polarization, since the dipoles induced in small spheres have a fixed orientation, causing them to scatter the wave intensively to another particle. This makes the optical interaction between them stronger, which leads to greater higher scattering. Moreover, it can be seen that the influences at *x*-polarized incidence are identical with those of *y*-polarized incidence. However, for the circularly polarized HOBB incidence, the scattering results of RCP and LCP wave are distinct, which does not correspond to the scattering results of isotropic spheres, since the scattering of isotropic spheres is symmetric with the RCP and LCP beam, which results in the complete coincidence. However, for chiral spheres, the inner field is decomposed into RCP and LCP components, corresponding to the different wave numbers, which results in the different scattering results.

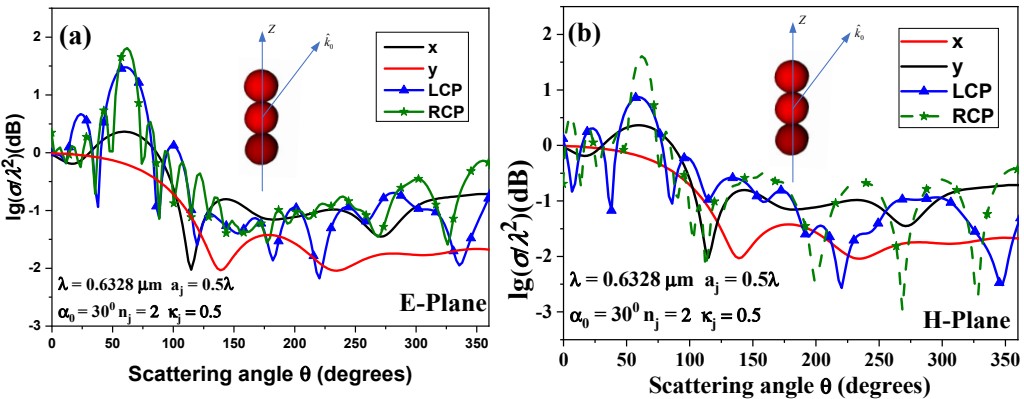

**Figure 4.** Angular distributions of normalized RCS induced by HOBB depending on the polarization states. (**a**) E-plane; (**b**) H-plane.

Figure 5 plots RCSs of a linear chain composed of three chiral particles by an RCP HOBB disseminating in the angle of 30° from the + Z axis. The three close-packed chiral particles are similar to Figure 4. The influence of beam conical angles on RCSs is shown in Figure 5a, and the related amplitude distributions are given in Figure 5b. If the beam order is immobilized as $p = 1$ and the conical angles of HOBB are changed to $\alpha_0 = 0°, 10°, 20°, 30°$, respectively, and if $p$ is fixed as constant, the center point size increases when the conical angle decreases. As given in Figure 5a, RCS decreases when we increase the conical $\alpha_0$; indeed, it is shown in Figure 5b that the bigger the angle $\alpha_0$, the lower the peak amplitude, which leads to a decrease in the amplitude of the RCS. Moreover, Figure 5a also demonstrates that the maximum RCS is influenced by both the conical angle and the angle of incidence, which always appears in the direction of the summation of these two angles. This is due to the effect that distinct half-conical angles can be treated as different planes of observation.

The beam order $p$ is a key parameter in the composition of HOBB, which strongly influence RCS. Figure 6 investigates the influences of RCS by an RCP HOBB disseminating in the angle of 30° from the + Z axis with different beam orders. The relevant parameters are identical to Figure 5. It is clear that a HOBB disseminates in the dark central area without spreading. Moreover, the central spot size depends highly on the beam order. The larger the beam order, the smaller the central spot size. To research the effect of the beam order, we fixed the beam conical angle $\alpha_0 = 30°$ and increased the beam order $p$ from 0 to 3, and the other parameters remained the same. Figure 6a depicts the calculated RCS. Figure 6b presents the corresponding amplitude distribution. Similarly, it can be observed that the maximum RCS is affected by the summation of the conical values and the angle of incidence in all cases.

As shown in Figure 6a, the related amplitude of RCS with respect to large beam orders are smaller than those of small beam orders, particularly for the maximum RCS positions. This is the result of an increase of beam order $p$, causing the decreasing intensity peaks as depicted in Figure 6b, resulting in the amplitude of RCS diminution. Moreover, it can be seen that if the order is not zero, HOBB appears with a hollow center in Figure 6b, which makes it minimal in the forward direction.

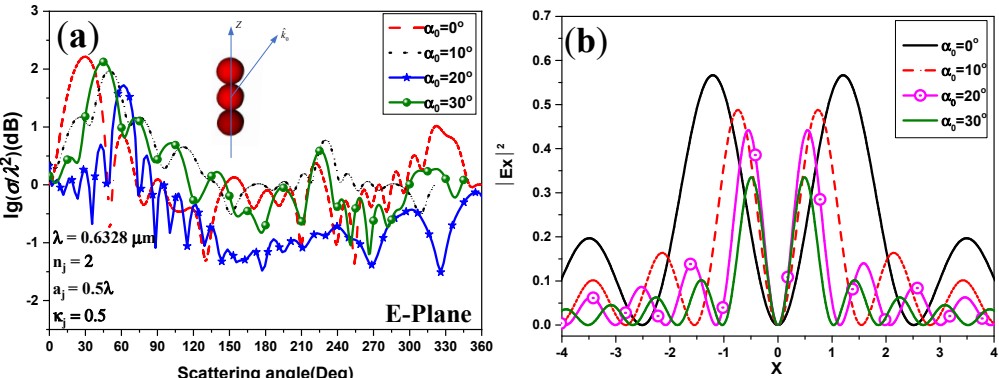

**Figure 5.** (**a**) Angular distributions of normalized RCS illuminated by an RCP HOBB based on the conical $\alpha_0$. (**b**) Distribution of a first-order Bessel beam with varying conical $\alpha_0$.

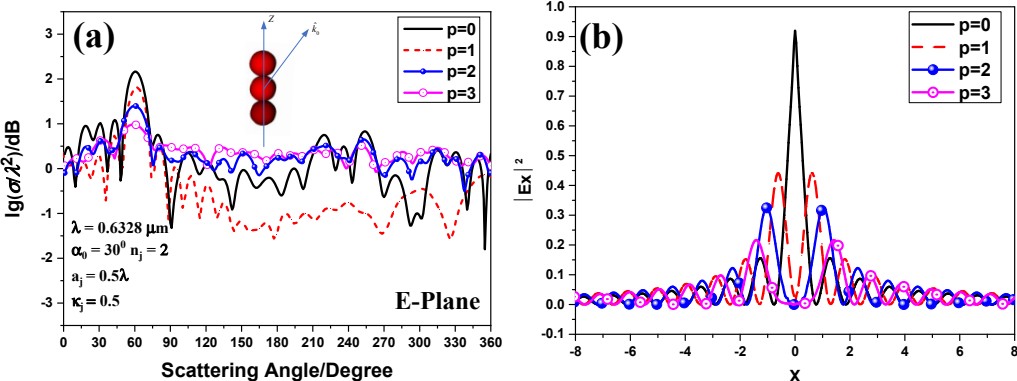

**Figure 6.** (**a**) Angular distributions of normalized RCS induced by an RCP polarized Bessel beam depending on the beam orders. (**b**) Distribution of a first-order Bessel beam with beam order $p$ as the parameter.

The RCS distributions of a linear chain composed of three close-packed chiral particles by a 632.8-nm first-order RCP Bessel beam propagating with varying incident angles from the +Z axis are shown in Figure 7. The relevant parameters are coincident with those shown in Figure 5. Selected simulations derived in Figure 6 show that the maximum RCS occurs in the direction of the combination of the angle of incidence and conical angles. For incident angle $\alpha = 90°$, considering that the chiral chain satisfies the symmetry distributions, the extreme position appears in the neighboring angle of maximum RCSs. This phenomenon indicates that the extremum point is more likely to occur in the adjacent direction of the maximum RCSs for vertical incidence, since the energy distribution is mainly focused on these two directions. In addition, for the case of oblique irradiation, RCS on the E- and H-planes no longer satisfies the symmetry distribution. Except for the circumstance of vertical incidence $\alpha = 90°$, the RCS in the H-plane is symmetrically distributed, since the symmetrical axe of the chiral chain is consistent with the direction of incidence; when the incident direction is in the E plane, the angular distributions of RCS in the H plane are symmetrically distributed.

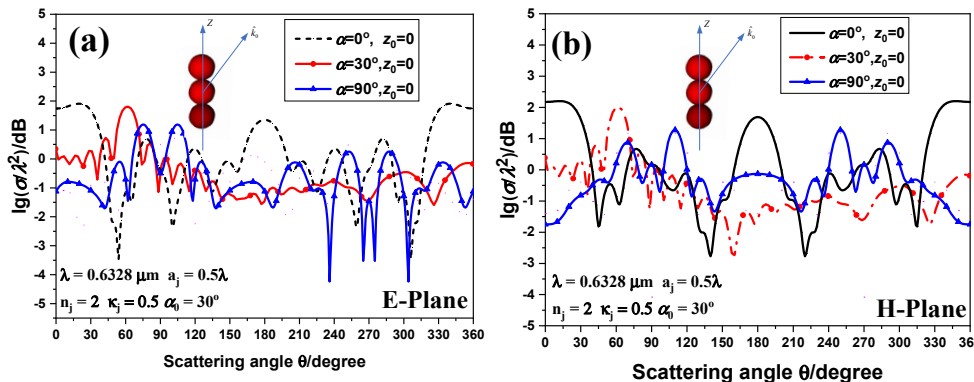

**Figure 7.** Angular distributions of normalized RCS illuminated by an RCP HOBB depending on the angles of incidence. (**a**) E-plane; (**b**) H-plane.

The RCS distributions of a linear chain composed of three close-packed chiral particles irradiated by a 632.8-nm Bessel beam are given in Figure 8. Under the same conditions as Figure 7, the influence of the varying chirality on RCS are considered, except for the case where the incident angle considers $\alpha = 30°$ from the $+ Z$ axis. The scattering enhances both in the E- and H-plane with the increase of chirality. Nevertheless, the angular distribution shape of RCS remains nearly unchanged. Therefore, the influence of the chirality on the extreme position can be ignored.

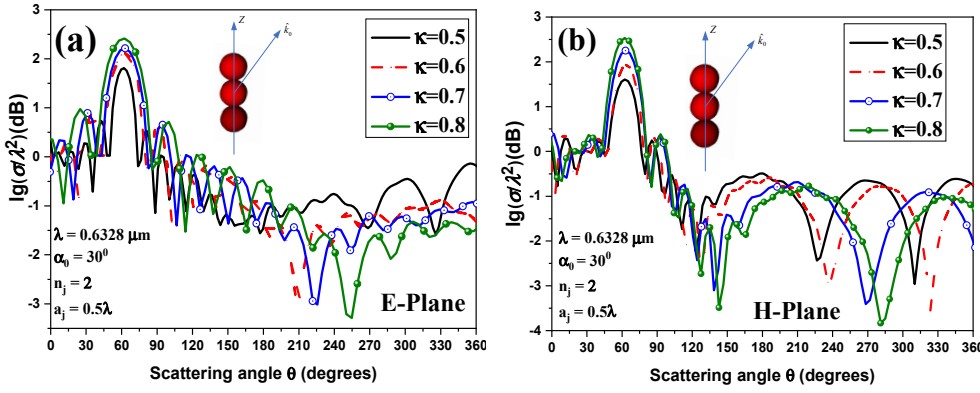

**Figure 8.** (**a**) Angular distributions of normalized RCS with varying chiral parameters $\kappa$ versus scattering angle irradiated by an RCP HOBB. (**a**) E-plane; (**b**) H-plane.

Figure 9 plots the RCS of a long linear chain consisting of 3, 7, 15, and 25 chiral spheres next to each other irradiated by RCP HOBB disseminating in the angle of 30° from the $+ Z$ axis. These chiral spheres are supposed to be the same with $\kappa_j = 0.5$, $a_j = 0.5\lambda$, and $\lambda = 0.6328 \,\mu$m. The centers of the HOBB and chiral chains are consistent. Figure 9 shows that the greater the number of chiral spheres that make up the structure, the nearer the extreme direction to the combination of the angle of incidence and conical angles. Moreover, when spherical numbers increase in the chiral structure, the overall amplitude of the RCS changes more greatly, and the vibrations become stronger. This is due to the result of more direct interactions between chiral particles.

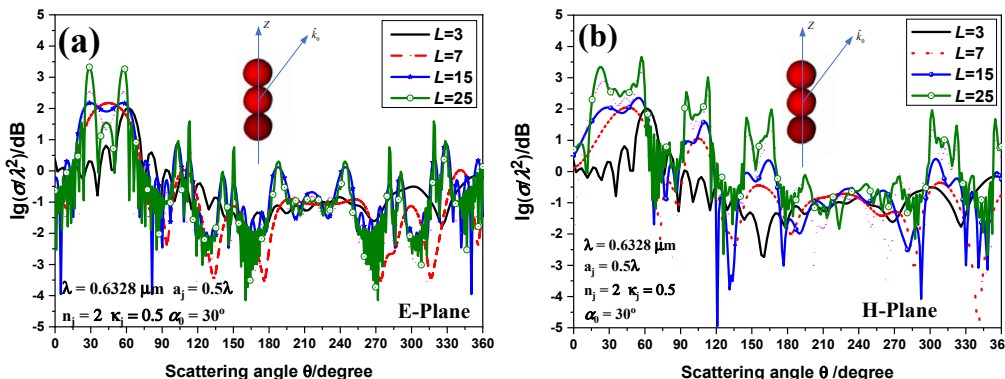

**Figure 9.** Angular distributions of normalized RCS with varying chiral spheres numbers versus scattering angle induced by an RCP polarized first-order Bessel beam. (**a**) E-plane; (**b**) H-plane.

In fact, the study of complex structure composed of a number of chiral particles may have some practical value. Therefore, we studied the multiple scattering of 12 identical chiral spheres arranged in $2 \times 2 \times 3$ cuboid lattice, depicted in Figure 10. The radii of every chiral particle is $a_j = 0.5\lambda$, and the periodic unit is $\lambda = 0.6328\ \mu$m. The array center is at the origin (0, 0, 0) in Oxyz.

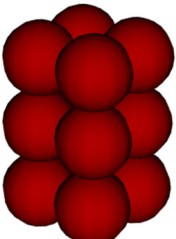

**Figure 10.** Illustration of a structure of assembled chiral particles composed in a $2 \times 2 \times 3$ square array.

The RCS of a $2 \times 2 \times 3$ array of 12 chiral spheres with different material losses induced by a 632.8-nm first-order Bessel beam propagating along the angle of $\alpha = 30°$ from the $+ Z$ axis is calculated in Figure 11. To research the effect of material loss, we chose the refractive index of the chiral sphere to be from $2.0 + 0.0i$ to $2.0 + 1.5i$, and kept the other parameters fixed. As given in Figure 10, for the unique structure of the chiral spheres, the extreme is affected by the combination of the angle of incidence and conical angles. With the increase of material loss, the shape of angular distribution remains nearly unchanged. Therefore, the effect of the refractive imaginary component on extreme position can be nearly ignored. Moreover, the corresponding RCS for great material loss is greater than small loss; the total influence of loss is to increase amplitude either in the E-plane or H-plane, which is attributed to the increased absorbance of photons, resulting in the effect of enhanced interactions on multiple scattering.

We investigated the scattering characteristics of other periodical arrays composed of *L* identical chiral particles. The periodical array presented in Figure 12 contains an $8 \times 8 \times 8$ cuboid array composed of 512 chiral particles. The related parameters are coincident with those shown in Figure 5, except that the centre is consistent with the underlying layer centre.

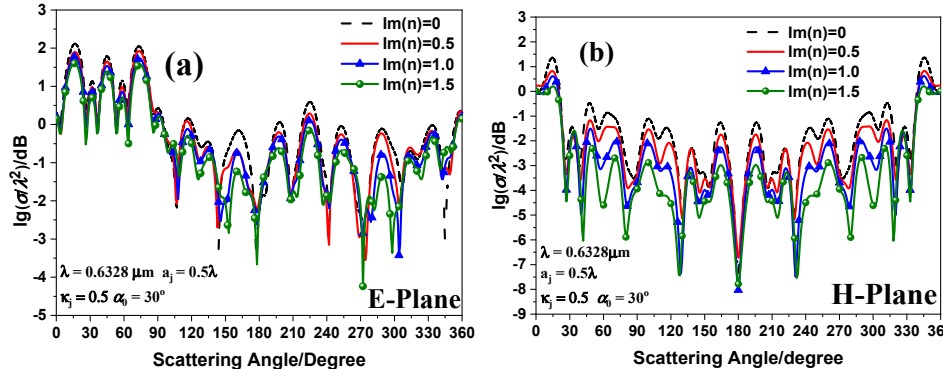

**Figure 11.** Angular distributions of normalized RCS with varying chiral spheres loss versus scattering angle induced by RCP polarized first-order Bessel beam. (**a**) E-plane; (**b**) H-plane.

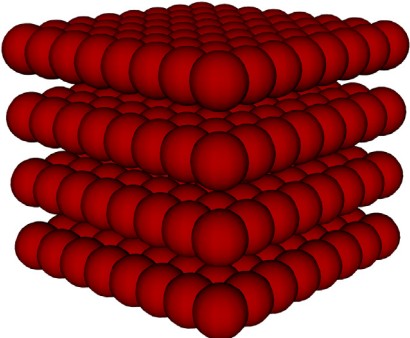

**Figure 12.** Illustration of a structure of 512 assembled chiral particles composed in an $8 \times 8 \times 8$ square array.

In Figure 13, the RCS distributions of the $1 \times 8 \times 8$ array of 64 chiral particles, $2 \times 8 \times 8$ array of 128 chiral particles, $3 \times 8 \times 8$ array of 192 chiral particles, and $4 \times 8 \times 8$ array of 512 chiral particles are investigated. The induced 632.8-nm Bessel beam propagates along the angle of $\alpha = 30°$ from the +Z axis. As shown in Figure 13, the extreme increases as the number of array layers increases. This result is mainly due to the denser multiple scattering. In addition, the RCS was found to be almost identical in the E- and H-plane in several directions. Moreover, the entire RCS decreases as the sphere number decreases; this is distinct from the previous characteristics investigated, as the chiral particles are thick. Moreover, for the complex periodical array of the chiral spheres, the extremum point disappears, denoting that the extreme is not only affected by the composed spherical number, but also affected by the structure of the chiral spheres.

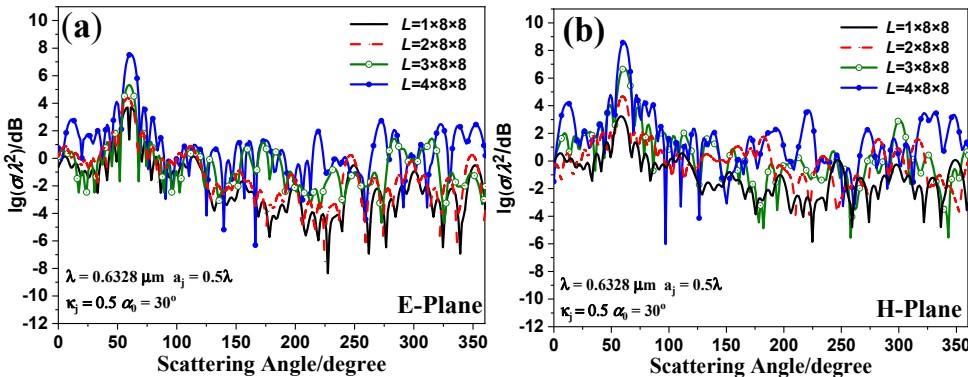

**Figure 13.** Angular distributions of normalized RCS of four structures composed of varying quantitative chiral particles versus the scattering angle from an RCP polarized first-order Bessel beam. (**a**) E-plane; (**b**) H-plane.

## 5. Conclusions

We investigate multiple interactions of collective chiral nanoparticles by an arbitrarily incident HOBB by using analytical solutions. The present theory and codes were proved to be effective by confrontation with the simulations obtained from the computer simulation technology (CST) software. Numerical results concerning the effects of beam order, beam conical angle, incident angles, beam polarization state, the chirality, the material loss on the scattering of various types of aggregated chiral particles, the number of linear chains, as well as the periodical structure with dense chiral spheres are displayed in detail. Results present that the interactional scattering of multiple chiral particles can be inherently distinct, depending both on the handedness of the chiral particles and the polarization directions. Therefore, when enhancing the interactions of multiple chiral particles, an appropriately polarized beam should be selected. Moreover, an extreme scattered value occurs in the combined direction of the angle of incidence and conical angles. Moreover, the amplitude of such a peak increases with the decrease of beam order and conical angle. With the increase of spherical numbers, the RCS distributions increase greatly. If plenty numbers of particles exist, RCSs of the E- and H-plane are almost identical in several propagating directions. Additionally, extreme position was indicated to be influenced by incident direction, the composed spherical linear chain number, and the structure of the spheres. The theoretical investigations provided here also support analytical investigation on the controlling of assembled chiral particles, which may find important applications in optical manipulation of aggregated chiral structure self-arrangement by focused Bessel beams.

**Author Contributions:** Conceptualization, J.B.; methodology, C.-X.G.; software, P.S.; validation, Y.G.; formal analysis, C.-X.G.; investigation, Z.-S.W.; writing—original draft preparation, J.B.; writing—review and editing, J.B.; funding acquisition, J.B. All authors have read and agreed to the published version of the manuscript.

**Funding:** This research was funded by the National Natural Science Foundation of China grant number 62001377.

**Institutional Review Board Statement:** Not applicable.

**Informed Consent Statement:** Not applicable.

**Acknowledgments:** The authors acknowledge the National Natural Science Foundation of China for its support of grant no. 62001377.

**Conflicts of Interest:** The authors declare no conflict of interest.

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
