# Peer review of "Light Interaction with Cluster Chiral Nanostructures by High-Order Bessel Beam"

_photonics, doi:10.3390/photonics9080509_

Round 1
Reviewer 1 Report
As far as I can understand, the authors want to carry out this work because previous research in scattering is only limited to a single chiral object, and the analytical study of interactions between aggregated chiral objects is very few. The focus of this paper is to report the interaction between the Higher-order Bessel beam and aggregated chiral objects, extending the study horizon beyond the Gaussian beam. The motivation for work is clear from the introduction section.
· However, since the Higher-order Bessel beam (HOBB) and chiral objects are two critical elements of this article, less space has been given to introducing Bessel beams. They should increase the references describing the importance and properties of the HOBB. Few suggestions are:
Recently, it has been reported that the HOBB, in non-ideal condition, reveals the orientation and topological charges of its helical wavefront, “Optics Communications Volume 485, 15 April 2021, 126710”, Also utilisation of the helical wavefront of HOBB for drilling “Phys. Rev. Applied 17, 034059 25 March 2022”.
· The authors didn’t explain fig. 2 clearly in the article. They should briefly explain what they were expecting and what they have got in fig.2 from the analysis.
· The authors report that…. “ As given in Fig.5 (a), RCS decreases when we increase the conical angle alpha0, which is satisfied with Fig.5 (b) that the bigger the angle alpha0, the lower the peak amplitude, which will lead to a decrease in the amplitude of the RCS”… However, in figure 5 (a), one can clearly see that for alpha0= 0 and 30 degrees, RCS has almost similar maximum amplitude (see the RCS for smaller values of scattering angle), which is contradictory. Please explain this behaviour.
Author Response
Dear professor,
Thanks so much for your constructive comments and suggestions on our manuscript. The detailed responses are below
Please see the attachment.

Reviewer 2 Report
In this paper, the authors described analytically, using series, the diffraction of a vector Bessel beam by several arbitrarily located chiral spheres. The proposed method was programmed and simulation was carried out using it. The correctness of the results was confirmed by comparison with the results of the commercial program CST. The results obtained in this work are new and can be published after the authors take into account the comments.
Comments
1. There are many borrowings from other works in the work. Equation (1) is taken from [36,42], equation (2) from [42], (4) from [42], (6) from [46], (13) from [47], (17) from [ 48], (19) from [49], (20)-(24) from [50], (26) from [51]. Therefore, the authors should explain what new they have brought to the theory of this method.
2. There are many uncertain values ​​in the work. The authors take equations from other papers and consider that the quantities in the equations need not be determined. The functions M and N in equation (2), the function P in equation (4), the constants "epsilon" and "mu" in line 167 are not defined. It is necessary to define all the designations used in the work.
3. Some abbreviations are given without explanation: CST, SVWF, RCP, LCP.
4. You should indicate the physical dimensions of the patterns in Fig.2. The numbers on the scales in Fig. 2 are not visible.
5. In equation (3) in the denominator, you must take the number m modulo, so that the denominator does not vanish.
6. Why in equations (10) and (11) the same factor from factorials is used twice? The denominator of this factor will vanish at s=n and s=-n.
7. Within what limits does the "sigma" in the sum (11) change?
8. References [32] and [43] are identical.
Author Response

(The authors gave the same response as above.)

Reviewer 3 Report
the presented numerical approach is consistent with CST modeling results. such cross check is very valuable and convincing. however, a better description is required to highlight possible benefits of new approach especially for computational implementation.
it would be useful to discuss how spin and orbital angular momenta can be implemented in the proposed approach. in many light manipulation experiments the light carries chirality, not only material.
Author Response

(The authors gave the same response as above.)
